**Data Availability Statement:** Due to the detailed nature of the transcripts and the small sample size

# Perspectives of people living with HIV on barriers to timely ART initiation following referral for antiretroviral therapy: A qualitative study at an urban HIV clinic in Kampala, Uganda

Micheal Kiyingi[1,2]*, Joaniter I. Nankabirwa[1,3], Christine Sekaggya Wiltshire[4], Joan Nangendo[3], John M. Kiweewa[5], Anne R. Katahoire[6], Fred C. Semitala[1,2,3]

1 Department of Medicine, Makerere University, Kampala, Uganda, 2 Makerere University Joint AIDS Program, Makerere University, Kampala, Uganda, 3 Infectious Diseases Research Collaboration, Kampala Uganda, 4 Infectious Diseases Institute, Makerere University, Kampala, Uganda, 5 Education Department, Fairfield University, Fairfield, Connecticut, United States of America, 6 Child Health Development Center, Makerere University, Kampala, Uganda

* michealkiyingi@gmail.com

## Abstract

Early initiation of antiretroviral therapy (ART) after HIV diagnosis prevents HIV transmission, progression of HIV to AIDS and improves quality of life. However, little is known about the barriers to timely ART initiation among patients who test HIV positive in settings different from where they will receive HIV treatment, hence are referred in the routine setting. Therefore, we explored the perspectives of people living with HIV on barriers faced to initiate ART following HIV testing and referral for treatment. In this qualitative study, we purposively sampled and enrolled 17 patients attending the Mulago ISS clinic. We selected patients (≥18 years) who previously were received as referrals for HIV treatment and had delayed ART initiation, as ascertained from their records. We conducted in-depth interviews, which were audio recorded, transcribed and translated. We used Atlas.ti version 9 software for data management. Data analysis followed thematic and framework analysis techniques and we adopted the socio-ecological model to categorize final themes. Key themes were found at organizational level including; negative experiences at the place of HIV diagnosis attributed to inadequate counselling and support, unclear communication of HIV-positive results and ambiguous referral procedures; and, long waiting time when patients reached the HIV clinic. At individual level, the themes identified were; immediate denial with late acceptance of HIV-positive results attributed to severe emotional and psychological distress at receiving results, fear of perceived side effects and long duration on ART. At interpersonal level, we found that anticipated and enacted stigma after HIV diagnosis resulted in non-disclosure, discrimination and lack of social support. We found that challenges at entry (during HIV test) and navigation of the HIV care system in addition to individual and interpersonal factors contributed to delayed ART initiation. Interventions during HIV testing would facilitate early ART initiation among patients referred for HIV care.

in this qualitative study, sharing such data can potentially identify the participants. The Makerere University School of Medicine research and ethics committee advises against sharing such information except on reasonable request to the corresponding author in consultation with the committee. Data is available upon request to the corresponding author and the School of Medicine Research and Ethics Committee, Makerere University via email (rresearch9@gmail.com) for researchers who meet the criteria for access to confidential data.

**Funding:** The Fogarty International Center (FIC), National Institute of Alcohol Abuse and Alcoholism (NIAAA), National Institute of Mental Health (NIMH), of the National Institutes of Health (NIH) under Award Number D43 TW011304, grant recipient: ARK; provided funding for this study. The funders had no role in the study design, data collection and analysis, decision to publish, or preparation of the manuscript. The content is solely the responsibility of the authors and does not necessarily represent the official views of the National Institutes of Health.

**Competing interests:** The authors have declared that no competing interests exist.

# Introduction

In 2016, Uganda adopted the World Health Organization (WHO) test and treat strategy, under which the current consolidated guidelines for human immunodeficiency virus (HIV) treatment, care, and support were developed. These guidelines recommend that patients who test HIV positive in a health facility or setting without anti-retroviral therapy (ART) services should be referred to another health facility with ART services and initiated on ART within 30 days from the positive HIV test [1,2]. These recommendations were intended to facilitate early ART initiation, which prevents HIV transmission, disease progression to acquired immunodeficiency syndrome (AIDS), and death among people living with HIV (PLHIV) [3–6], and, is an important step towards achieving the global target to end AIDS by 2030 [7]. However, continued delay to initiate ART poses a challenge in sub-Saharan Africa (SSA) [8–11], and many patients still present with advanced HIV disease [12,13]. Previous studies have identified multiple barriers to ART initiation among PLHIV attending HIV clinics or participating in research settings [14–17], but there are limited studies among patients who are referred for HIV care in the routine setting. In addition to individual and social factors, these patients may face added challenges to navigate the health system, due to the unavailability of ART at some of the HIV testing points, lack of harmonized referral guidelines and inadequate follow up.

In Uganda, HIV testing is available in most health facilities (including public and private health facilities) and in the community (during community outreach and self-tests) [18]. After testing HIV positive, a patient is linked to a HIV clinic where they will initiate ART. The linkage requires a referral if the testing setting has no ART services. Free ART services (at no cost to the patient) are only available at HIV clinics located in public and private-not-for profit health facilities. The referral practice varies from place to place due to lack of national referral guidelines. Fortunately, standard referral notes are available [19] and newly diagnosed PLHIV can access ART medicines at no cost from any HIV clinic with or without referral documentation. However, despite knowing their HIV status, some patients may delay to visit the HIV clinic to initiate ART, which increases the risk of HIV transmission and poor clinical outcomes. A 2021 study conducted by our team among referred adult PLHIV found that 15.4% delayed to initiate ART. We also found that having been diagnosed with HIV from a private health facility, initial denial of HIV positive results and not receiving a follow up phone call from health workers after referral were all significantly associated with delayed ART initiation [20]. Therefore, as a follow up study, we sought to explore the perspectives of PLHIV on barriers to timely ART initiation at a tertiary urban HIV clinic following HIV diagnosis from elsewhere.

# Methods

## Design, setting and participants

This qualitative study was conducted at Mulago Immune Suppressive Syndrome (Mulago ISS) clinic in Kampala—Uganda. This is the largest HIV clinic in Uganda serving a population of over 16,500 patients and is a center of excellence for comprehensive HIV and Tuberculosis services. The community served is mainly urban and peri-urban, living in and around the capital city, Kampala. The services at this clinic are supported by the Makerere University Joint AIDS Program (MJAP) and are offered at no cost to the patient.

Study participants were patients who presented to Mulago ISS clinic from 1st to 24th May 2021 for HIV care. The first author and three experienced research assistants (one male and two females) purposively sampled and enrolled eligible patients who were attending a routine clinical visit. Those included were: 1) 18 years and older, 2) had their HIV diagnosis made

outside Mulago ISS clinic, 3) visited Mulago ISS clinic and initiated on ART between February—May 2021 (within 3 months prior to study contact), and 4) found to have delayed ART initiation as per national guidelines (more than 30 days elapsing since HIV diagnosis) [21]. We conducted this study in accordance with the declaration of Helsinki [22], and ethical approval was obtained from the School of Medicine Research and Ethics Committee (SOMREC), Makerere University College of Health Sciences, approval number REC REF 2020–120. All participants provided written informed consent prior to enrolment in the study.

## Data collection and analysis

We conducted in-depth interviews at the Mulago ISS clinic among eligible patients using an interviewer administered in-depth interview guide. The interview guide was developed based on previous studies on barriers to ART in community settings conducted in East Africa and similar settings. The interview guide was pretested at another HIV clinic in Kampala and appropriate adjustments made. The interviews were conducted by the two research assistants who were social scientists with experience in conducting qualitative research in an HIV setting. To ensure privacy and encourage free expression among participants, interviews were conducted in quiet private counselling rooms at the Mulago ISS clinic. As the interviews proceeded, participants were identified by purposive sampling using the maximum variation strategy, where we varied them based on age group [(18–24 (5), 25–40 (9), > 40 (3)], sex (Males—8 and Females—9) and place of HIV diagnosis (9 from private health facility, 6 from public health facility and 2 during community outreach). The interviews explored participants' subjective experiences on testing HIV positive, referral for ART initiation, as well as reasons for delaying ART initiation and their motivation to initiate ART. Each interview lasted for 45–60 minutes, conducted in the patient's language of preference (Luganda or English) and responses recorded on a voice recorder. Saturation was achieved after 17 interviews, where, no new information was collected and we could derive categories from the data after iterative preliminary data review and analysis.

Data analysis was guided by thematic and framework analysis techniques [23,24]. We used Atlas.ti version 9 software [25] for data management. Recorded data was transcribed verbatim and all transcripts translated to English by two independent translators. The translated transcripts were then back translated into Luganda by the two interviewers who had conducted the original interviews. Consensus was reached during weekly meetings with the first author. The research team reviewed the transcripts iteratively to assess for accuracy and allow familiarization with the data. We then highlighted the main points in the transcripts, grouped them, which groups we identified by codes. We identified the main themes that emerged from the data, grouped the related sub-themes and summarised the meaning of each sub-theme in a condensed form. The developed codes were grounded in the data and revisions in response to emerging patterns were made. The table that characterises the themes, sub-themes, codes and their respective frequencies is uploaded as S2 File.

To guide generation and naming of themes, we drew upon empirical research findings from a 2018 study that established barriers to ART among PLHIV participating in a community level cluster randomized test and treat trial in rural East Africa [15]. The themes identified in that study included; HIV associated stigma manifested as denial and fear of disclosure, poverty, lack of social support, negative prior experiences with health services and drug side effects. We then mapped the themes on to the socio-ecological model [26] which provides an understanding of the health behaviors and the environment in which they occur [27], and, establishes the inter-relationships between the individual, the social, the physical and policy environment [28]. After a positive HIV test, the ability to continue navigating the health

system is very important for patients who test in settings without ART services, and individual, interpersonal and organizational factors can all contribute.

## Results

We conducted seventeen (17) in-depth interviews. More than half of the participants, 9 (52.9%) were female, between age 25–40 years and 12 (70.6%) had dependents. Most participants, 9 (52.9%) had their HIV diagnosis made from a private health facility prior to attending the Mulago ISS clinic for ART initiation (Table 1).

We developed themes at three levels of the socio-ecological model, namely;

### 1. Individual barriers to timely ART initiation

At the time of receiving the HIV positive results, patients reported having experienced fear manifesting as emotional and psychological distress followed by immediate denial of results. They reported having cried, fainted and needed time to come to terms with the diagnosis of HIV. Some sentiments captured were;

*"I would have come (to initiate ART) but I was scared, I was crying and I even became unconscious."* **(Male, 25 years)**.

*"I did not accept the results, I cried and they tried counseling me but I refused. I asked her [health worker] where are you taking me, are they going to test me again? Because it is possible for the results to be different."* **(Female, 30 years)**

The acceptance process took a long time and the deterioration in physical health pushed patients to seek treatment at the HIV clinic. Sometimes this happened after trying out ineffective treatments. The deteriorating health status drew them from the initial state of denial into acceptance of the HIV positive status as a reality. A patient described his experience as;

**Table 1. Socio-demographic characteristics of study participants.**

| Demographic | Characteristic | Number (n = 17) | Proportion (%) |
|---|---|---|---|
| Sex | Male | 8 | 47.1 |
| | Female | 9 | 52.9 |
| Age | 18–24 | 5 | 29.4 |
| | 25–40 | 9 | 52.9 |
| | 41 and above | 3 | 17.6 |
| Highest level of education attained | None or Primary | 11 | 64.7 |
| | Secondary | 5 | 29.4 |
| | Tertiary | 1 | 5.9 |
| Marital status | Single/ living with no partner | 10 | 58.8 |
| | Married/ living with partner | 7 | 41.2 |
| Has dependents | Yes | 12 | 70.6 |
| | No | 5 | 29.4 |
| Employment status | Self–employed | 10 | 58.8 |
| | Formal employment | 3 | 17.6 |
| | Not employed | 4 | 23.5 |
| Place of HIV diagnosis | Public health facility | 6 | 35.3 |
| | Private Health facility | 9 | 52.9 |
| | Community outreach | 2 | 11.8 |

"...what exactly brought me was that I was dying off, I had a disease that was killing me, the money had gotten done. I had to come to the doctors to help me." **(Male, 42 years).**

Another patient described her process of acceptance as:

"Now it's like this, by the time I decide to come here at the hospital (HIV clinic), I have given up and whatever you tell me is what I will go by; not that I will start saying 'the machine didn't work well', or 'the machine was faulty' or anything of the sort, no, I just accepted." **(Female, 27 years)**

## 2. Interpersonal barriers to timely ART initiation

Patients expressed fear of stigma and discrimination if they enrolled into ART programs in the communities where they lived. They reported feeling paranoid regarding what community members would say about their HIV status in their absence, with many concealing their HIV status from family and friends. One patient expressed the prevailing sentiment regarding disclosure when she noted;

"It is true that I am HIV positive, but how will it be [people knowing that I am HIV positive]? How will people take it? Even now I don't want people to know that I am positive." **(Female, 35 years).**

The fear of stigma resulted in non-disclosure, which was coupled with limited support from partners, family members and friends, and anticipating having to hide ART medicines that delayed initiation of ART. In response to whether there was disclosure to any friends or relatives, a 23-year-old female patient retorted;

"No, because I was afraid. Moreover, my friends shouldn't know because it is my life, not theirs." **(Female, 23 years).**

Another patient commented on anticipating having to hide ART medicines from her family;

"The fear for my side was to take that medication without my children knowing." **(Female, 44 years).**

Other concerns identified were anticipated side effects and duration of ART, which participants had heard from other people regarding ART. Some of such sentiments were put this way;

"Some people were scaring me that it (ART) makes one run mad and does a lot of things to someone. Even myself, one day it did a lot of things to me." **(Male, 25 years.)**

"I heard people say that you have to take the medications every day without fail, so that is what I feared and scared me the most." **(Male, 44 years).**

Patients mentioned concerns they had regarding side effects of ART that they had heard or seen other people on ART experience, which had deterred them from initiating ART. They reported changes of skin, drugs too powerful for their body, inability to walk, and body weakness. One 27-year-old female respondent described having seen her sister get drug side effects;

*"Now for her [her sister], when she took the medication, she got a lot of skin rashes and I was also scared of that. . .But when I started on the medication, I did not get any problem."* **(Female, 27 years).**

### 3. Challenges at organizational level that hindered timely ART initiation

Patients reported to have experienced challenges at the health facility where they tested HIV positive, especially at privately owned health facilities. The challenges included having received inadequate counselling and support during HIV testing. One patient put it this way;

*"They did not give me any support. They just scared me and told me that I was HIV positive, and I should go to Mulago main hospital to confirm."* **(Female, 35 years).**

Patients also commented on the testing and referral procedures at the health facilities where they tested HIV positive that did not give clear HIV test results, ambiguous referral instructions and no referral notes given to them. A respondent narrated;

*". . . I had come to Kampala to work. I went to that company and they interviewed me, then they took my blood sample. Then, the health worker was like, 'we don't understand you, whether you are positive or negative.' I was like, okay. Time passed and I had to come here to know the truth. I told them I want to go abroad to work and that's when they tested me and told me that I was HIV positive."* **(Female, 28 years).**

Another patient who also had an HIV test done at a private health facility before travel for work described her experience as;

*"I was slated to go abroad to work with some company, but I did not know that I was HIV positive. When the company did not get back to me, I came to this clinic and asked to be tested."* **(Female, 28 years).**

At the HIV clinic, reported barriers included long waiting time and the large number of clients. Some patients attributed the long waiting time to the many initial procedures for enrollment of the new clients. One patient elaborated;

*"At times we could come early and leave late, and so I decided to arrive late morning. Many times, I found people who had come early complaining because they had not been attended to yet."* **(Female, 44 years).**

## Discussion

In this study, we present patient perspectives on barriers to timely ART initiation among PLHIV referred to Mulago ISS clinic for HIV treatment in the routine care setting. We found multiple barriers at individual and interpersonal levels during and immediately after testing HIV positive, which is a key entry point in to the HIV care continuum [29], and, challenges at organizational level that hindered patients from easily navigating the HIV care system to access ART. Our results show that for referred PLHIV, the circumstances around testing HIV positive are key and can potentially delay ART initiation.

We found key barriers at organizational level, mainly the facilities where patients had had their HIV diagnosis. Patients complained of inadequate counselling and support during HIV testing, unclear communication of HIV positive results and ambiguous referral procedures at

the site of HIV diagnosis especially at private health facilities. An HIV test is the entry point into the HIV care continuum. If the HIV test is negative, the client is linked to HIV prevention services, while if positive, they are linked to HIV care and treatment services [21,29]. The barriers we found at this level not only make entry and navigation of the HIV care system difficult and delay ART initiation, but may also contribute to the barriers we found at both individual and interpersonal levels. Previous studies revealed that individualized counselling and clear referral procedures including follow up phone calls by the healthcare workers at the HIV testing site helped patients initiate ART [14,30–32]. A quick decision to initiate ART by the patient at this point cannot be over-emphasized and needs close attention by HIV testing service providers. Training and support of the providers of HIV testing services helps to empower them to support patients to cope with their diagnosis, decide to initiate ART and navigate the health system more easily [33,34].

At individual level, patients reported barriers related to emotional and psychological distress at receiving HIV positive results that led to immediate denial of results with acceptance attributed to deterioration of physical health condition. In addition, they reported concerns they had about ART medicines including side effects, size of pills and duration of ART. These may be due to the inadequate counselling and poor communication of HIV results we found at organizational level above. Individualized counselling at this stage would help patients decide to initiate ART early. The long acceptance process has negative implications on the community HIV transmission, ART adherence and clinical outcomes [4,35,36] and supporting these patients to accept the results immediately after testing HIV positive would help them to decide to initiate ART early [37].

At interpersonal level, patients complained of anticipated stigma resulting in non–disclosure, worries about the side effects of ART medicines they had heard or seen on other people and anticipating to hide ART medicines from family members. Previous studies have identified similar barriers that hinder ART initiation in East Africa and similar settings including denial of HIV results, fear of drug side effects and stigma among others [9,15,38–42]. Community perceptions still play a crucial role for PLHIV to initiate ART and interventions to encourage disclosure, improve family support and fight stigma can potentially improve timely ART initiation.

## Limitations

This study focused on PLHIV received as referrals at a large urban HIV clinic. As a result, experiences of PLHIV referred to lower level HIV clinics were not considered yet they constitute a large combined population of PLHIV in Uganda. However, both large and small HIV clinics use the same HIV prevention and treatment guidelines. We propose a larger study including more HIV clinics of variable sizes and geographical locations to comprehensively understand the barriers to timely ART in this population.

The perspectives of health workers at the HIV testing centers, caregivers of the patients, community members and policy experts were not included in this study. Inclusion of these populations would broaden the understanding of barriers to ART among referred patients to facilitate development of comprehensive interventions.

The patients had to recall the events that happened when they tested HIV positive and referred for ART, which could have caused a recall bias. However, we minimized this by reducing the eligibility period to within 3 months since ART initiation.

## Conclusion

This study identified a complex interplay of barriers to timely ART initiation at individual, interpersonal and organizational levels. In addition to known individual and interpersonal

barriers to ART, entry and navigation of the HIV care system at organizational level was a major challenge for referred HIV positive patients.

## Supporting information

**S1 File. COREQ checklist.**
(PDF)

**S2 File. Codebook.**
(PDF)

## Acknowledgments

The authors acknowledge the staff and patients of Mulago ISS clinic where the study was conducted. The authors also acknowledge the "Strengthening behavioral and social science research capacity to address evolving challenges in HIV care and prevention in Uganda" project at Makerere University, which provided technical support for this study through training and mentorship. This project aims to strengthen capacity in behavioral and social sciences research to address evolving challenges in HIV in Uganda.

## Author Contributions

**Conceptualization:** Micheal Kiyingi, Joaniter I. Nankabirwa, Christine Sekaggya Wiltshire, Anne R. Katahoire, Fred C. Semitala.

**Data curation:** Micheal Kiyingi.

**Formal analysis:** Micheal Kiyingi, Christine Sekaggya Wiltshire, Joan Nangendo, John M. Kiweewa, Anne R. Katahoire, Fred C. Semitala.

**Funding acquisition:** Micheal Kiyingi, Joan Nangendo, Anne R. Katahoire, Fred C. Semitala.

**Investigation:** Micheal Kiyingi, Joaniter I. Nankabirwa, Christine Sekaggya Wiltshire.

**Methodology:** Micheal Kiyingi, Christine Sekaggya Wiltshire, John M. Kiweewa, Anne R. Katahoire, Fred C. Semitala.

**Project administration:** Micheal Kiyingi, Joaniter I. Nankabirwa, Joan Nangendo, Fred C. Semitala.

**Resources:** Micheal Kiyingi, Anne R. Katahoire, Fred C. Semitala.

**Software:** Micheal Kiyingi.

**Supervision:** Micheal Kiyingi, Christine Sekaggya Wiltshire, Anne R. Katahoire, Fred C. Semitala.

**Validation:** Micheal Kiyingi, John M. Kiweewa, Anne R. Katahoire.

**Visualization:** Micheal Kiyingi, Joan Nangendo, John M. Kiweewa, Fred C. Semitala.

**Writing – original draft:** Micheal Kiyingi.

**Writing – review & editing:** Micheal Kiyingi, Joan Nangendo, John M. Kiweewa, Anne R. Katahoire, Fred C. Semitala.

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
