## [Decision Letter · Decision Letter 0]

20 Feb 2023

PGPH-D-22-01999

Perspectives of people living with HIV on barriers to timely ART initiation following referral for antiretroviral therapy: A qualitative study at an urban HIV clinic in Kampala - Uganda.

Dear Dr. Michael Kiyingi,

Thank you for submitting your manuscript to PLOS Global Public Health. After careful consideration, we feel that it has merit but does not fully meet PLOS Global Public Health’s publication criteria as it currently stands. Therefore, we invite you to submit a revised version of the manuscript that addresses the points raised during the review process.

We look forward to receiving your revised manuscript.

Kind regards,

Ferdinand Mukumbang, PhD

Academic Editor

Journal Requirements:

1. Please send a completed 'Competing Interests' statement, including any COIs declared by your co-authors. If you have no competing interests to declare, please state "The authors have declared that no competing interests exist". Otherwise please declare all competing interests beginning with the statement "I have read the journal's policy and the authors of this manuscript have the following competing interests:"

b. If any authors received a salary from any of your funders, please state which authors and which funders.

Additional Editor Comments (if provided):

Reviewers' comments:

Reviewer's Responses to Questions

**Comments to the Author**

1. Does this manuscript meet PLOS Global Public Health’s publication criteria? Is the manuscript technically sound, and do the data support the conclusions? The manuscript must describe methodologically and ethically rigorous research with conclusions that are appropriately drawn based on the data presented.

Reviewer #1: Partly

Reviewer #2: Partly

2. Has the statistical analysis been performed appropriately and rigorously?

Reviewer #1: Yes

Reviewer #2: N/A

3. Have the authors made all data underlying the findings in their manuscript fully available (please refer to the Data Availability Statement at the start of the manuscript PDF file)?

Reviewer #1: Yes

Reviewer #2: Yes

4. Is the manuscript presented in an intelligible fashion and written in standard English?

Reviewer #1: Yes

Reviewer #2: No

5. Review Comments to the Author

Reviewer #1: Thank you for the opportunity to review this manuscript. Delayed ART initiation and nonadherence to HIV treatment are major public health issues, particularly in sub-Saharan Africa. While the study addressed a significant problem in HIV care engagement, the manuscript however, needs major revisions in the methods, results, and discussion sections for it to be publishable. I have detailed some concerns that the authors need to address below.

Abstract

Line 41: Add information about the location of the study.

Line 42: Specify the age range of study participants in parenthesis after “adult patients”

Background

Line 63 : Change “A” in “Anti-retroviral therapy” to a small letter “a”

Line 63 : Change “P and L” in “People Living” to a small letters.

Line 76: What does “flexible” HIV testing mean? Either hash out what it means or simply remove it from the sentence.

Line 78 : Change “an HIV” to “a HIV”

Line 84/85: Revise phrasing “some patients may delay to present to the HIV clinic” for clarity. Please specify if you mean patients delay going to the clinic to initiate ART or if they delay clinic visits in general.

Methods

Line 104: I understand the meaning in context of “presented to Mulago” but I suggest that you revise this to “visited Mulago”, which is a terminology that is commonly used in HIV/health research.

Data collection

Line 112-119: This section needs improvement, the description of data collection procedures is unclear. The authors mentioned in the previous section that purposive sampling was used to recruit participants into the study, however did not explain/describe how. Please clarify where interviews were conducted (e.g. at clinic, community, etc) and by whom?

Results

Line 141/142: Please provide a table showing socio-demographic characteristics of study participants. This information is very important for context and helps other researchers/readers to better understand the population of interest and interpretation of study findings.

In addition, since findings in both the abstract and data analysis sections are being presented and interpreted according to the socio-ecological model (SEM). For consistency, authors need to revise/reorganize the themes in the results section to reflect the SEM framework i.e. individual level, interpersonal, organizational, etc.

Line 177: Revise “report to the ART clinic” and use clearer phrasing e.g. “seek treatment/medical care”.

Line 183-186: add quotation marks to participant’s response.

Line 202: Revise descriptors for consistency. Pick one format (Female, 35 years) or (Female PLHIV, 23 years) and revise all the other descriptors throughout this section.

Discussion

Line 242: It is unclear which themes have been identified at the interpersonal level. Here authors’ discuss findings “at individual and interpersonal levels” yet in previous sections, all 5 themes were classified as either “at the individual level or organizational level”. Please revise the results section and clarify which themes were identified at the individual, interpersonal, and organizational levels.

Line 250: Revise sentence and change “he/she is” to “they are”

Line 253: Revise “individual and interpersonal levels”, see comment above.

Line 260: Revise “At intrapersonal and interpersonal levels”

Line 267: Provide a reference (s) to support this statement “Important to note, for the population we studied, the risk of loss to follow up after testing HIV positive is quite high…”

Conclusions

Line 290-291: This conclusion is not entirely supported by the data presented. Please see comment above regarding categorization of themes using the SEM.

Reviewer #2: I found major technical issues pertaining to various sections of this manuscript:

1. sampling vs. population were not clearly explained. How did the study reached the saturated number of informants?

2. Instrument: the semi-structured questions were developed or adopted from which study / literature?

3. need to clearly describe the back translation process as this involve more than one language. This step is crucial to ensure the authenticity of the original expressed language with the intended meaning conveyed during the translation.

4. what is the saturation point for the informants?

5. The main literature use to frame this study was an old study conducted in 1988. Whereby in today's society, social media disrupt and create a whole set of challenges to health communication.The study did not mention this and provide satisfactory comparative analysis to validate their reason/s of using an old study as point of reference.

6. When analysis was conducted using Atlas ti, surely the presentation of result would have fineness in terms of table, abstraction processes, table of frequency, etc.

7. The results were too shallow and did not succinctly provide a manifest and latent analysis of the prescribed themes.

8. Introduction to results was too simple.

9. If the extraction of quotations / analysis were done using Atlas ti, the source would be identified in more detail manners with the inclusion of percentile, minutes of the quotation, and discourse unit.

10. Out of 38 references, only 21 were latest five years studies. The rest were old. Recommend for revise and add latest citations.

11. Some citations did not follow the required citation style. Revise.

12. This manuscript should be sent for proofread for cohesion and clarity.

6. PLOS authors have the option to publish the peer review history of their article (what does this mean?). If published, this will include your full peer review and any attached files.

**Do you want your identity to be public for this peer review?** For information about this choice, including consent withdrawal, please see our Privacy Policy.

Reviewer #1: No

Reviewer #2: **Yes: **Suffian Hadi Ayub

---

## [Decision Letter · Decision Letter 1]

21 Jun 2023

Perspectives of people living with HIV on barriers to timely ART initiation following referral for antiretroviral therapy: A qualitative study at an urban HIV clinic in Kampala - Uganda.

PGPH-D-22-01999R1

Dear Dr. kiyingi,

We are pleased to inform you that your manuscript 'Perspectives of people living with HIV on barriers to timely ART initiation following referral for antiretroviral therapy: A qualitative study at an urban HIV clinic in Kampala - Uganda.' has been provisionally accepted for publication in PLOS Global Public Health.

Best regards,

Julia Robinson

Executive Editor

Reviewer Comments (if any, and for reference):

Reviewer's Responses to Questions

**Comments to the Author**

1. If the authors have adequately addressed your comments raised in a previous round of review and you feel that this manuscript is now acceptable for publication, you may indicate that here to bypass the “Comments to the Author” section, enter your conflict of interest statement in the “Confidential to Editor” section, and submit your "Accept" recommendation.

Reviewer #1: All comments have been addressed

2. Does this manuscript meet PLOS Global Public Health’s publication criteria? Is the manuscript technically sound, and do the data support the conclusions? The manuscript must describe methodologically and ethically rigorous research with conclusions that are appropriately drawn based on the data presented.

Reviewer #1: (No Response)

3. Has the statistical analysis been performed appropriately and rigorously?

Reviewer #1: Yes

4. Have the authors made all data underlying the findings in their manuscript fully available (please refer to the Data Availability Statement at the start of the manuscript PDF file)?

Reviewer #1: Yes

5. Is the manuscript presented in an intelligible fashion and written in standard English?

Reviewer #1: Yes

6. Review Comments to the Author

Reviewer #1: Line 42: Is ISS an acronym for something? If so, spell it out since it is being used for the first time here.

7. PLOS authors have the option to publish the peer review history of their article (what does this mean?). If published, this will include your full peer review and any attached files.

**Do you want your identity to be public for this peer review?** For information about this choice, including consent withdrawal, please see our Privacy Policy.

Reviewer #1: No
